# Recent Advances of Cell-Penetrating Peptides and Their Application as Vectors for Delivery of Peptide and Protein-Based Cargo Molecules

**DOI:** 10.3390/pharmaceutics15082093

**Published:** 2023-08-07

**Authors:** Huifeng Zhang, Yanfei Zhang, Chuang Zhang, Huan Yu, Yinghui Ma, Zhengqiang Li, Nianqiu Shi

**Affiliations:** 1School of Pharmacy, Jilin Medical University, Jilin 132013, China; hfjlmu@163.com (H.Z.); zyf20181812@163.com (Y.Z.); jlu_cz@hotmail.com (C.Z.); yuhuanjlu@163.com (H.Y.); mayinghui04329@163.com (Y.M.); 2Key Laboratory for Molecular Enzymology and Engineering of the Ministry of Education, College of Life Sciences, Jilin University, Changchun 130012, China; lzq@jlu.edu.cn; 3College of Pharmaceutical Sciences, Yanbian University, Yanji 133002, China

**Keywords:** CPPs, peptides, proteins, discovery of CPPs, new generation of CPPs, cargo delivery

## Abstract

Peptides and proteins, two important classes of biomacromolecules, play important roles in the biopharmaceuticals field. As compared with traditional drugs based on small molecules, peptide- and protein-based drugs offer several advantages, although most cannot traverse the cell membrane, a natural barrier that prevents biomacromolecules from directly entering cells. However, drug delivery via cell-penetrating peptides (CPPs) is increasingly replacing traditional approaches that mediate biomacromolecular cellular uptake, due to CPPs’ superior safety and efficiency as drug delivery vehicles. In this review, we describe the discovery of CPPs, recent developments in CPP design, and recent advances in CPP applications for enhanced cellular delivery of peptide- and protein-based drugs. First, we discuss the discovery of natural CPPs in snake, bee, and spider venom. Second, we describe several synthetic types of CPPs, such as cyclic CPPs, glycosylated CPPs, and D-form CPPs. Finally, we summarize and discuss cell membrane permeability characteristics and therapeutic applications of different CPPs when used as vehicles to deliver peptides and proteins to cells, as assessed using various preclinical disease models. Ultimately, this review provides an overview of recent advances in CPP development with relevance to applications related to the therapeutic delivery of biomacromolecular drugs to alleviate diverse diseases.

## 1. Introduction

With recent developments in the field of life sciences, numerous biologically active drugs have emerged in the therapeutic drug market. One-third of quasi-innovative drugs approved by the U.S. Food and Drug Administration (FDA) in recent years have been biomacromolecules or biopharmaceuticals, with a proportion that is increasing each year [1]. Peptides and proteins are two important classes of biomacromolecules with roles in biopharmaceutical applications. Peptides, which are defined as polypeptide chains containing 2–50 amino acids, have molecular weights of less than 5000 Da. Importantly, polypeptides possess a high degree of secondary structure but lack tertiary structure, with both secondary and tertiary structures enabling biological functions [2].

Peptide-based drugs were initially developed to serve as hormone analogs to correct hormonal imbalance-related disorders. Nowadays, peptides play numerous roles in bioprocesses that cannot be fulfilled by small molecules [3]. As compared with traditional small molecule-based drugs, peptide- and protein-based drugs provide several key advantages. For example, they exert minimal off-target effects [4], possess low toxicity, possess low immunogenicity and antigenicity, are easily modified, possess high bioactivity, and can be used in various forms (vaccines, antibodies, and hormones) [5]. However, peptides have limitations, such as their lack of receptor selectivity, easy denaturation, tendencies to undergo aggregation and/or adsorption, short half-life, low oral bioavailability, and high production costs [2]. Regardless, it should be noted that both peptides and proteins cannot readily penetrate the cellular plasma membrane lipid bilayer to reach intracellular targets, due to their relatively large molecular sizes and intrinsic physicochemical properties. Consequently, strategies must be devised to overcome this obstacle to facilitate the delivery of peptides and proteins to the intracellular environment.

Many physical, biological, and chemical methods have been explored to address this challenge. One such method entails the use of cell-penetrating peptides (CPPs), which have increasingly replaced traditional cellular drug delivery approaches by enabling safer, more efficient direct transport of biomacromolecules across cell membranes both in vitro and in vivo. CPPs, also referred to as protein transduction domains (PTDs), comprise a class of short peptides that commonly contain 5–30 amino acids (most of which are positively charged amino acids, such as lysine and arginine) [6]. The first synthetic CPP that was developed, the TAT peptide, was derived from the human immunodeficiency virus in 1988 [7,8]. Thereafter, the number of reported CPPs derived from natural proteins or synthesized derivatives has increased rapidly, such that the current number of CPPs is unknown, although the CPPsite2.0 CPP database contains 1855 entries. Each entry provides extensive information about a particular CPP that includes: (i) peptide sequence, (ii) nature of peptide, (iii) chemical modifications, (iv) experimental validation techniques, (v) structure of peptides, and (vi) type of cargo delivered [9].

Due to the high number of different CPPs in existence, numerous CPP classification schemes have been proposed. One traditional scheme assigns CPPs to three classes based on their physical and chemical properties: cationic, amphipathic, and hydrophobic. Alternatively, CPPs have also been categorized according to origin into protein-derived, chimeric, and synthetic classes [10], while yet another CPP classification scheme is based on cell-specific and non-cell-specific characteristics of certain clinically important CPPs. Nevertheless, the abovementioned distinct classification schemes, which cover a broad range of CPPs but often overlap, should be replaced by a single, standardized scheme to simplify CPP classification. Towards this end, many new classification approaches have been proposed, such as linear vs. cyclic, predicted vs. random, etc. [6]. However, further discussion of proposed schemes is needed before a suitable standardized scheme can be developed.

Regardless of CPP classification challenges, CPPs are undoubtedly highly promising and convenient agents for use in clinical applications. Although only about 30 years have passed since the discovery of the first CPP, several pharmaceutical companies have begun to develop CPPs for clinical applications. Currently more than 25 CPP-based products are under clinical evaluation, including a few under evaluation in Phase III clinical trials [11]. Nevertheless, only one anionic and amphipathic antitumor CPP derived from bacterial azurin p28 peptide, which targets solid and central nervous system (CNS) tumors expressing p53, has reached Phase I clinical trial status (NCT00914914) [12].

In this review, we provide an overview describing CPP discovery, several new types of CPPs, and recent research on CPP applications for the intracellular delivery of peptides and proteins. In the first section, which focuses on CPP discovery, CPPs in snake, bee, and spider venoms are discussed. In the second section, several new types of CPPs are highlighted, such as cyclic CPPs, glycosylated CPPs, and D-form CPPs. In the third and final section, cell membrane permeability and therapeutic applications of different CPPs that have been developed to deliver peptides and proteins to cells in preclinical models of various diseases are summarized and discussed. Ultimately, this review provides an overview of recent advances in the development of CPPs for use in the delivery of peptide- and protein-based therapeutics to treat diverse diseases.

## 2. Discovery of CPPs

To date, the discovery of highly efficient CPPs has been challenging, and there are commonly three approaches to finding new CPPs. One approach entails CPP design and synthesis based on specific amino acids, motifs, peptides, or other types of modifiable constituents. The second approach has identified some cell tissue-specific CPPs for drug delivery using phage display biopanning, such as MT23 (LPKQKRRQRRRM), tLyP-1 (CGNKRTR), and so on. The third approach is based on the discovery of CPPs present in natural sources, including viruses, microorganisms, plants, animals, and so on. For example, gH625 (HGLASTLTRWAHYNALIRAF), a membranotropic peptide, was derived from herpes simplex virus type I by Stefania Galdiero’s group [13] and utilized in delivering self-assembled peptide-based nanofibers (NFs) with a matrix metalloproteinase-9 (MMP-9)-responsive sequence to enhance the penetration efficiency and tumor-triggered cleavage of doxorubicin in triple-negative breast cancer cells [14]. Recently, an increasing number of CPPs derived from animal toxins and venom peptides have been discovered. In this section, we present examples of CPPs that have been obtained from snake, bee, and spider venoms. For additional approaches related to CPP design and synthesis, the reader can refer to a recent excellent review in [15].

### 2.1. CPPs from Snake Venom

Decades ago, crotamine, a low molecular weight cationic membrane active peptide [16], was first isolated from the venom of *Argentinean rattlesnakes* [17]. Subsequently, crotamine (Table 1) was found to consist of forty-two amino acid residues, including nine lysine, two arginine, and six cysteine residues that are interconnected via three disulfide bridges [18]. Crotamine is a versatile molecule with an immunomodulatory activity that is rapidly internalized by cells. More recently, researchers exploring clinical applications of crotamine via biodistribution studies have reported that crotamine antitumor activity requires homing of the molecule to tumor cell nuclei [19], while other researchers have reported crotamine antimicrobial activities [16]. Even though crotamine has promise as a clinically useful agent, local and systemic acute inflammatory responses can be triggered by intradermal injection of the natural form of the peptide into rats, which has limited its therapeutic applicability [17]. To address this issue, since 2020, crotamine-based cell-penetrating nanocarriers have been developed as delivery systems for use in introducing nucleic acids into cells and animals. Among them, plasmid DNA–crotamine nanoparticles (NPs) and crotamine-functionalized gold NPs can selectively penetrate rapidly proliferating cells of diverse organisms, including mice, Plasmodium, and worms. Moreover, crotamine–silica NPs have been shown to provide slow crotamine delivery that enables therapeutic administration of decreased doses that provide the same therapeutic benefit as higher doses, but with fewer adverse effects. In addition, these crotamine nanocarriers are effective even when used to orally administer the agent [20]. Meanwhile, nucleolar targeting peptides (NrTPs) (Table 1) and cytosol-localizing peptides (CyLoPs), which are miniature crotamine mimics, have been developed. These mimics include CyLoP-1 (Table 1), a membrane-active peptide-based cargo delivery vector used for both mammalian and plant cells, which also exerts antimicrobial activity against methicillin-resistant *Staphylococcus aureus*. Notably, cysteine residues of CyLoPs play major roles in the abovementioned activities, as evidenced by marked decreases in these activities when cysteines are replaced with serine residues [21]. NrTPs, another family of CPPs, can successfully penetrate various cell types (except for erythrocytes) and thus have been used as drug delivery vehicles with remarkable nucleolus-homing properties, even at low concentrations. Among them, NrTP6 provides the best penetration efficiency with no cytotoxicity, while internalized NrTP7 and NrTP8 may be cytotoxic [22]. Furthermore, NrTP6 can deliver a near-infrared dye to cells when it is coupled to the dye via its free N-terminal cysteine residue. In this form, the modified NrTP6 can serve as a potent probe that can be adapted for use in image-guided applications for pinpointing heterogeneity of tumors for targeting heterogenic subpopulations [23]. Additionally, crotalicidin (Ctn) and its C-terminal fragment Ctn (Table 1), which are cationic peptides derived from rattlesnake venom, possess cell-penetrating antimicrobial, antifungal, and antitumor properties. These agents exhibit cell selectivity, accumulate within cytoplasmic membranes and nuclei of cancer cells, and predominantly enter cells via direct entry that depends on endocytic peptide uptake processes [24].

### 2.2. CPPs from Bee Venom

As shown in Table 1, melittin (MEL), a main component of bee venom, is a 26-residue linear, water-soluble, cationic, hemolytic, and amphipathic peptide [25]. MEL has been reported to exert various biological effects, due to its antiviral, anti-inflammatory, anticancer, antidiabetic, anti-infective, adjuvant, and delivery capabilities. These activities all depend on non-selective cytolysis that can disrupt both prokaryotic and eukaryotic cell membranes [26]. Hybrid peptides were designed based on the incorporation of proapoptotic peptide d (KLAKLAK)_2_ (dKLA) and MEL specifically bound to M_2_-like tumor-associated macrophages (TAMs) to induce apoptosis of CD206^+^ M_2_-like TAMs after cell membrane penetration with no cytotoxic effect [26]. By contrast, MEL is a non-selective cytolytic peptide that disrupts membranes of all prokaryotic and eukaryotic cells and thus exhibits a high degree of toxicity with associated side effects (e.g., hemolysis). These issues have prompted researchers to investigate other MEL delivery vehicles, such as liposomes [27], lipid disks [28], and micelles [29]. Among them, Ce6/MEL@SAB was designed based on incorporation of serum albumin (SA)-coated boehmite (B), an aluminum hydroxide oxide, to generate an organic–inorganic scaffold that could be loaded with photosensitizer chlorin e6 (Ce6) and MEL. Notably, the use of this system to treat tumors in mice demonstrated that laser irradiation combined with a single injection of Ce6/MEL@SAB wiped out more than 30 percent of tumors in vivo, while significantly reducing hemolytic side effects elicited by MEL treatment alone [30].

Apamin (Table 1), another main peptide derived from bee venom, is composed of 18 amino acids and exerts many biological effects, including anti-inflammatory and antinociceptive effects, as well as cytotoxic effects against cancer cells. In addition, apamin selectively inhibits Ca^2+^-activated K^+^ channel functions and blood–brain barrier (BBB) penetration functions without causing inflammation [31,32]. Moreover, apamin is remarkably stable when exposed to diverse pH and temperature conditions and is resistant to serum protease degradation [33]. However, as apamin has been reported to traverse the BBB to exert its neuroprotective effects, the agent is neurotoxic and can induce convulsions when administered at high doses. To address this issue, non-toxic apamin analogs such as ApOO ([Orn13,14] apamin) (Table 1) have been synthesized [34] and subsequently found to act as non-toxic BBB-penetrating peptides that remained stable when tested using an in vitro cell-based BBB model and when dissolved in human serum and injected intravenously into human subjects [31]. Therefore, ApOO is a promising CPP for use in delivering drugs to cells which has led to its inclusion in several new strategies to deliver antitumor drugs to tumor cells. For example, a self-assembling NP was successfully created by grafting a 12-mer D-enantiomeric peptide (DPMIβ) onto apamin. This NP exhibited a favorable pharmaceutical profile when it was assessed for passive tumor targeting, cell membrane penetration, intracellular reductive response, and endosome escape activities [35].

### 2.3. CPPs from Spider Venom

Latarcins include seven short linear antimicrobial and cytolytic peptides with potential cell-penetrating activities that were purified from the venom of the spider species *Lachesana tarabaevi* [36]. These peptides, which possess amphipathic alpha-helical structures, exhibit general cytotoxic activity that is responsible for their reported antifungal, antimicrobial, hemolytic, and anticancer effects [37].

Lycosin-I (Table 1), a linear, cationic, 24-residue CPP with high selectivity for tumor cells, has been found in the venom of spiders of the species *Lycosa singoriensis*. This CPP can inhibit cell proliferation and induce tumor cell apoptosis by activating the mitochondrial death pathway and by triggering an increase in tumor cell p27 level. However, the fact that this CPP does not penetrate the cell plasma membrane efficiently has limited its application as an antitumor agent for use in treating solid tumors. To address this issue, researchers have modified lycosin-I via amino acid substitution, which has led to the creation of substituted lycosin-I CPPs with increased anticancer activities relative to that of lycosin-I. One such CPP, R-lycosin-I, is a cytotoxic peptide derived from lycosin-I through the replacement of all Lys residues with Arg residues that have endowed this CPP with significantly different biological activities, secondary structure, hydrodynamic size, and zeta potential as compared to its lycosin-I parent [38]. Furthermore, R-lycosin-I (Table 1) penetrated deeper into 3D tumor spheroids and exhibited stronger tumor inhibitory activity and greater serum stability than lycosin-I when incubated with solid tumor cells [39].

**Table 1 pharmaceutics-15-02093-t001:** CPPs derived from animal toxins and venom peptides.

Name	Peptide Sequence	Origin	Ref.
Crotamine	AASSSGGPPPGGGGGCCCCCMILT PPTTLLLLLLLLLHHAATAV	The venom of Argentinean rattlesnakes	[18]
CyLoP-1	CRWRWKCCKK	Crotamine mimicry	[21]
NrTP1	YKQCHKKGGKKGSG	Crotamine mimicries	[22,23]
NrTP2	YKQCHKKGG-Ahx-KKGSG		
NrTP5	ykqchkkGGkkGsG		
NrTP6	YKQSHKKGGKKGSG		
NrTP7	YRQSHRRGGRRGSG		
NrTP8	WKQSHKKGGKKGSG		
Crotalicidin (Ctn)	KRFKKFFKKVKKSVKKRLK KIFKKPMVIGVTIPF-NH_2_	Rattlesnake venom	[24]
Ctn[15–34]	KKRLKKIFKKPMVIGVTIPF-NH_2_		
RhB-Ctn	RhB-KRFKKFFKKVKKSVKKRL KKIFKKPMVIGVTIPF-NH_2_		
CF-Ctn	CF-KRFKKFFKKVKKSVKKR LKKIFKKPMVIGVTIPF-NH_2_		
b-Ahx-Ctn	b-Ahx-KRFKKFFKKVKKSVK KRLKKIFKKPMVIGVTIPF-NH_2_		
RhB-Ctn[17–36]	RhB-KKRLKKIFKKPMVIGVTIPF-NH_2_		
Ctn[17–36]-Lys(RhB)	KKRLKKIFKKPMVIGVTIPFK(RhB)-NH_2_		
CF-Ctn[17–36]	CF-KKRLKKIFKKPMVIGVTIPF-NH_2_		
BF-Ctn[17–36]	BF-KKRLKKIFKKPMVIGVTIPF-NH_2_		
b-Ahx-Ctn[17–36]	b-Ahx-KKRLKKIFKKPMVIGVTIPF-NH_2_		
Melittin (MEL)	GIGAVLKVLTTGLPALISWIKRKRQQ-NH_2_	European honeybee Apis mellifera venom	[37]
Apamin	CNCKAPETALCARRCQQH	Japanese solitary wasp Anoplius samariensis venom	[28]
ApOO([Orn13,14] apamin	LOHATGPACL	Apamin mimicry	[32]
Latarcin-1	SMWSGMWRRKLKKLRNALKKKLKGE	Spider Lachesana tarabaevi	[25]
Lycosin-I	GKLQAFLAKMKEIAAQTL-NH_2_	Spider Lycosa singorensis	[40]
R-lycosin-I	GRLQAFLARMREIAAQTL-NH_2_	An arginine modification of lycosin-I in which all Lys residues were replaced by Arg residues	[27,41]

## 3. New Generations of CPPs

Although CPPs have excellent capabilities when used for gene and peptide/protein delivery, liposome functionalization, and NPs, they also have many drawbacks, such as potential toxicity, low selectivity, and susceptibility to protease degradation and endolysosomal entrapment [42]. In order to overcome these limitations and enhance CPPs’ suitability for clinical therapeutic applications, several new generations of CPPs with optimized properties have been developed via different techniques, such as cyclization, glycosylation and chirality alterations, etc. Below, we describe several technical advances related to the generation of new types of CPPs for diverse applications.

### 3.1. Cyclic CPPs

CPPs have been used to deliver various cargoes, such as peptides, proteins, nucleic acids, and NPs, into mammalian cells both in vitro and in vivo. However, these CPPs, which are mainly linear CPPs, possess certain limitations that have hampered their use in clinical settings [43], such as endosomal entrapment, toxicity, poor cell specificity [44], susceptibility to proteolytic degradation [45], suboptimal cell penetration, and very poor cytosolic delivery efficiency (<5%) [46].

To overcome these limitations, several researchers have investigated cyclization as a potentially useful strategy to reduce CPPs’ conformational freedom, while greatly enhancing their metabolic stabilities and binding affinities/specificities for target molecules [47]. Meanwhile, other researchers have also reported that the cyclization of CPPs may improve their cellular entry efficiencies. For example, the results of one study demonstrated that cyclic arginine-rich CPPs, such as cyclic Tat and R_9_, exhibited enhanced cellular uptake kinetics relative to their linear and more flexible counterparts. This effect was attributed to cyclization-induced increases in distances between arginine residues [48]. Moreover, the delivery efficiency of another cyclic heptapeptide, cFΦR4 (Φ is l-2-naphthylalanine), when used to bring cargo into the cytoplasm and nucleus, was 4–12-fold higher than Tat, penetratin, and nonarginine. This remarkable effect was attributed to the ability of cFΦR4 to directly bind to membrane phospholipids to facilitate their endocytic internalization by human cancer cells and their subsequent escape from entrapment in early endosomes within the cytoplasm [49]. Furthermore, other researchers also discovered that cyclic CPPs were efficient transporters of functional full-length proteins, such as green fluorescent protein (GFP), into live cells, while uncyclized CPPs carrying similarly large cargoes were endocytically taken up but remained trapped in endocytic vesicles [50].

As expected, cyclic CPPs possess greater stability when exposed to proteases than linear CPPs, prompting researchers to rationally design cyclic CPPs for use in transdermal and orally bioavailable cargo delivery. For example, the cationic cyclic CPP TD-34 (ACSSKKSKHCG), which was designed based on the sequence of TD-1 (ACSSSPSKHCG) using the partial arginine or lysine scan method, was found to support greater percutaneous insulin absorption, transmembrane penetration ability, and proteolytic stability as compared to uncyclized TD-1 [51,52]. More recently, a family of small, amphipathic cyclic peptides with greater resistance to proteolytic degradation has been generated [53]. The most striking of these are CPP1 [cyclo(FΦRRRRQ)] and CPP9 [cyclo(fΦRrRrQ)], due to their excellent stabilities that support oral bioavailability. In particular, several amphipathic homochiral L-CPPs also possess nuclear-targeting properties. For example, cyclic [WR]_4_- and cyclic [CR]_4_-mediated 2–5-fold greater human ovarian adenocarcinoma (SK-OV-3) cellular uptake of (F’)-Gly-(pTyr)-Glu-Glu-Ile (F’-GpYEEI) as compared to that mediated by unconjugated F’-GpYEEI. In addition, fluorescently labeled phosphopeptide was predominantly translocated into nuclei after it formed a complex with cyclic [WR]_5_ peptides, while no cellular uptake of un-complexed, labeled phosphopeptide was observed [54]. In another study, cyclic CPPs containing [WR]_4_ were found to efficiently transport various hydrophilic and hydrophobic molecules. For example, when paclitaxel (PTX) and camptothecin (CPT) were conjugated to the cyclic peptide [W(WR)_4_K(βAla)], all conjugates exhibited improved water solubility and stronger cytotoxicity against various cell lines [55]. Furthermore, in contrast to doxorubicin (Dox), Dox conjugated to [C(WR)_4_K] localized to the nuclei of four cancer cell lines and exhibited significantly reduced toxicity when added to mouse myoblast cells at the same concentration [55]. The structures of cyclic CPPs mentioned above are shown in Figure 1.

Cyclic CPPs include monocyclic (e.g., all of the abovementioned cyclic CPPs), bicyclic, and tricyclic CPPs. All cyclic CPPs contain arginine and other amino acids, such as tryptophan and cysteine residues. Bicyclic peptides, a unique group of peptides that are characterized by two intramolecular linkages [56], have a more constrained conformation, reduced flexibility, improved binding properties, greater proteolytic stability, and more robust cellular uptake ability for cargo delivery than monocyclic CPPs [57]. Concurrently, [W_5_G]-(triazole)-[KR_5_] and [W_5_E]-(β-Ala)-[KR_5_] bicyclic peptides composed of tryptophan and arginine residues have been shown to significantly enhance cellular delivery of F’-GpYEEI into nuclear and cytosolic compartments of SK-OV-3 cells [58]. This enhanced activity was attributed to the fact that the linker between the two monocyclic components was a flexible amide or non-flexible triazole, such that the amide linkage within c[W_5_E]-(β-Ala)-c[KR_5_] may have contributed to the greater observed cellular uptake ability of this negatively charged phosphopeptide. Therefore, these results demonstrated that an optimal balance between a positive charge and hydrophobicity is needed to support CPP interactions with the cell membrane to achieve deep CPP penetration of the lipid bilayer [47]. Meanwhile, arginine-rich bicyclic peptides synthesized using cysteine perfluoroarylation chemistry followed by covalent linkage to phosphonodiamidite morpholino oligonucleotide (PMO) have been shown to deliver PMO to cells to treat a genetic disease caused by exon-skipping activity [59]. As an alternative strategy, one research group evaluated a new type of bicyclic peptide composed of two rings that perform separate functions, whereby one ring is a CPP ring that enables cellular entry and the other ring binds to the target protein [60]. Using this strategy, they created a large combinatorial library of cell-permeable bicyclic peptides that included peptide 7, a bicyclic named Tm (GWIYA)Δ(FΦRrRΔ)-BBK (Figure 2) that contained 3–6 random residues in the A ring and 12 different CPP sequences consisting of phenylalanine or naphthylalanine and arginine residues in the B ring. These bicyclics performed dual functions of cell penetration and target engagement and could efficiently enter cytosolic compartments of mammalian cells. Once they entered the cytosol, they selectively inhibited the canonical NF-κB signaling pathway (a family of transcription factors with key roles in regulating the immune response, inflammation, cell differentiation, and cell survival) by blocking the NF-κB essential modulator (NEMO)–IKKβ interaction without altering non-canonical NF-κB pathway activity [61]. However, like any drug delivery system, cyclic CPPs also have their limitations and defects, such as limited cargo delivering capacity, non-targeted delivery, relatively complicated synthesis, rapid clearance, immunogenicity, and degradation by proteases. Moreover, these bicyclics also selectively inhibited proliferation of cisplatin-resistant ovarian cancer cells, as evidenced by their low μM IC_50_ values.

Recently, research groups have synthesized tricyclic peptides, including a trimer of monocyclic peptide [WR]_4_ named [WR]_4_-[WR]_4_-[WR]_4_ [62]. This tricyclic enhanced cellular uptake of fluorescently labeled phosphopeptide (F’-GpYEEI), anti-HIV drugs (e.g., lamivudine, emtricitabine, stavudine), and siRNA by cells of the breast cancer cell line MDA-MB-231. Moreover, this compound also exhibited modest antibacterial activity against methicillin-resistant Gram-positive bacteria. Using a different approach, another research group designed and synthesized two Tat trimers, designated tri-Tat A and tri-cTat A for linear and cyclic forms of Tat, respectively. They then covalently linked these trimeric peptides to an azide-functionalized scaffold, designated scaffold A (Figure 3A). Similarly, two other Tat trimers, designated tri-Tat B and tri-cTat B (for linear and cyclic Tat trimers, respectively), were linked to an alkyne-functionalized scaffold designated scaffold B (Figure 3D). As compared with monomers and linear Tat trimers, the two tricyclic CPPs, tri-cTat A and tri-cTat B, provided significantly more potent delivery of functional antibodies and antibody fragments across the plasma membrane into cytosols of cancer cells, while avoiding endosomal entrapment of delivered biomacromolecules [63]. Thus, beneficial properties of cyclic CPPs, such as efficient delivery of diverse types of cargoes, nuclear localization, proteolytic stability, low cytotoxicity, and endocytosis-independent cellular uptake, make them promising next-generation pharmaceutical delivery vehicles [64].

### 3.2. Glycosylated CPPs (GCPPs)

Despite the excellent delivery capabilities of CPPs when used for nucleic acid and protein delivery, liposomal functionalization, and NP applications, CPPs have several limitations, such as potential toxicity and lack of selectivity [42]. Nevertheless, as compared to the administration of insulin alone, nasal co-administration of insulin and penetratin in rats yielded 50% bioavailability and achieved significant blood glucose reduction [65]. More recently, glycosylation of CPPs has emerged as a promising new strategy for use in increasing CPP target cell selectivity and reducing toxicity to non-target cells. For example, (R_6_/W3), a glycosylated CPP (GCPP) with the sequence Ac-RRWWRRWRR-NH_2_, was synthesized with one, two, or three galactose residues in place of tryptophan indole side-chains that were incorporated within the peptide via triazole linkages to produce four GCPPs (Table 2). These GCPPs were then coupled to the proapoptotic KLAK peptide (KLAKLAKKLAKLAK) via a disulfide bridge. As compared to non-glycosylated conjugates, such as (R_6_/W3) S-S-KLAK and penetrating S-S-KLAK, the four GCPPs exhibited significantly decreased cytotoxicity towards wild-type Chinese hamster ovary (CHO) cells, while also decreasing internalization of the KLAK cargo (Table 2). These results indicated that GCPPs succeeded in increasing KLAK peptide cell selectivity, while reducing KLAK toxicity towards non-target cells [66]. In 2011, the same team found that the introduction of galactosyl residues between central residues within CPPs’ amphipathic secondary structures reduced CPPs’ internalization efficiencies except when galactosyl residues were linked N-termini of CPPs that contained tryptophan residues. Moreover, the viability of CHO cells was negatively correlated with number of galactosyl moieties [67].

**Table 2 pharmaceutics-15-02093-t002:** Representative GCPPs in this article.

Name	Peptides and Primary Sequences	Ref.
(R_6_/W2)-[Pra-Gal(OH)]-SH	Ac-C-RRW-[Pra(Gal-OH)]-RRWRR-NH_2_	[66]
(R_6_/W0)-[Pra-Gal(OH)]_3_-SH	Ac-C-RR-[Pra(Gal-OH)]_2_-RR-[Pra(Gal-OH)]-RR-NH_2_	[66]
(R_6_/W2)-[Pra-Gal(CH_2_)_3_(OH)]-SH	Ac-C-RRW-[Pra-Gal(CH_2_)_3_(OH)]-RRWRR-NH_2_	[66]
(R_6_/W2)-[Pra-Gal(CH_2_)_3_(OAc)]-SH	Ac-C-RRW-[Pra-Gal(CH_2_)_3_(OAc)]-RRWRR-NH_2_	[66]

Pra propargylglycine, Pra-Gal(OH) β-D-galactopyranoside triazole, Pra-Gal(CH_2_)_3_(OH)_3_ (β-D–galactopyranosyl-oxy)propyl triazole, Pra-Gal(CH_2_)_3_ (OAc) 2,3,4,6-tetra-O-acetyl-Pra-Gal(CH_2_)_3_(OH).

Importantly, the abovementioned introductions of glycan residues into the peptide structure usually reduced CPPs’ internalization efficiency and increased their cellular toxicities towards different cell lines. To address these issues, another research group utilized a new strategy involving the functionalization of CPPs through incorporation of glycan residues via oxime bond-based linkages. They next synthesized a series of GCPPs containing an amphiphilic helical peptide scaffold with the amino acid sequence RRLLRRLKRL, then incorporated two reactive alkoxyamines as connectors that were functionalized via oxime bonding-mediated additions of glycan residues, including β-D-galactose, α-D-mannose, N-acetyl-β-D-glucosamine, β-D-glucose, and a branched trisaccharide aldehyde of α-D-mannose (Figure 4A) to generate CPPs designated TmP(Gal)_2_, TmP(Man)_2_, TmP(NAG)_2_, TmP(Glu)_2_, and TmP(Man_3_)_2_, respectively. In order to minimize the impact of incorporated alkoxyamine pendants on the CPP secondary structure, alkoxyamine pendants that were inserted at the interphase were designed to contain segregated cationic and hydrophobic domains (Figure 4B) [68]. Taken together, the aforementioned results indicate that glycans conjugated to a helical peptide scaffold could modulate peptide scaffold cell-penetrating behavior and efficiency. As compared to the TmP(Acetone)_2_ control peptide, the series of GCPPs exhibited much lower toxicity and slightly lower penetrating efficiency that could be improved without increasing toxicity by slightly increasing the peptide concentration. Therefore, glycosylation of a helical peptide scaffold may be used to improve the balance between GCPPs’ uptake efficiency and toxicity. In conclusion, although GCPPs hold promise, numerous issues must be addressed before GCPPs can serve as next-generation drug delivery tools.

### 3.3. D-Form CPPs

In the 1970s, D-amino acids such as D-alanine (D-Ala), D-glutamate (D-Glu), and D-aspartate (D-Asp) were first discovered in nature. Since that time, they have been experimentally shown to play important roles in numerous organisms. Although D- and L-amino acids have similar physicochemical properties, their physiological functions are completely different [69]. For example, several researchers have reported that replacement of L-amino acids with D-amino acids can increase resistance of proteins to enzymatic proteolysis [70] (Table 3). Consequently, D-amino acids exhibit longer gastrointestinal, plasma, and intracellular half-lives than L-amino acids [71]. Therefore, to protect CPPs from premature degradation, D-amino acids have been incorporated as substitutes for corresponding L-amino acids. For example, the half-life of pVEC, a CPP derived from the murine vascular endothelial cadherin protein, was 10.5 min in phosphate buffer containing 10 units of trypsin and 44.6 min in phosphate buffer containing 4.2 units of carboxypeptidase A and 18 units of carboxypeptidase B. By contrast, its counterpart, which consisted solely of corresponding D-amino acids, remained intact in serum and resisted enzymatic degradation [72]. This strategy has also been used to produce other CPP D-amino acid counterparts with enhanced proteolytic resistance, such as Tat [73], R_9_ [74], penetratin [75], RICK [76], and sweet arrow peptide (SAP) [77]. Notably, increased serum stability is not only limited to CPPs solely comprising D-amino acids, but is also a characteristic of CPPs that contain both D- and L-amino acids (tpTPTGTQtpt), such that both types of CPPs exhibit high-level resistance to proteolytic degradation in human serum and lysosomes [78].

**Figure 4 pharmaceutics-15-02093-f004:**
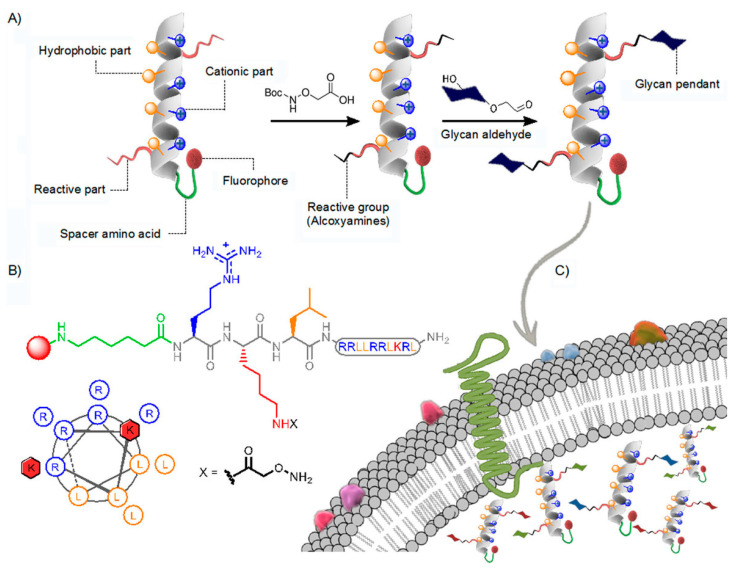
Peptide structure and membrane transport. (**A**) The peptide segregated domains along an alpha-helical scaffold. (**B**) Linear peptide primary sequence and heptad-based wheel diagram for the helical conformation of the peptide. (**C**) Membrane translocation and cellular internalization are modulated by the glycan residue. Reproduced with permission from reference [70].

Uptake efficiency differences between D- and L-amino acid-containing CPPs have been reported to vary greatly, with some reports indicating that cellular uptake of CPPs such as polyarginine, Tat peptide, and penetratin were not influenced by amino acid chirality [73,79], while other reports indicated that uptake efficiency was enhanced for D-amino acids-containing CPPs, due to their increased proteolytic stability [79]. However, results obtained by Wouter P.R. and colleagues [80] refuted the earlier paradigm, because the abovementioned studies were conducted on a rather limited set of cell types. They then used different cell lines to comprehensively investigate factors underlying reported differences in cellular uptake of nonarginine (R_9_), penetratin, human lactoferrin-derived peptide hLF (38–59), and their counterparts solely comprising D-amino acids. Their results revealed more efficient cellular uptake of cationic L-CPPs as compared to that of their D-CPP counterparts when these CPPs were added to HeLa and MC57 fibrosarcoma cells, but not when they were added to Jurkat cells. Differences in results obtained for different cell lines were attributed to the fact that heparan sulfate (HS) is present within plasma membranes of Jurkat cells but not in plasma membranes of HeLa and MC57 cells, while L- and D-peptides exhibited similar binding affinities to HS. Thus, cellular binding of CPPs to HS did not depend on chirality, while endocytic uptake of CPPs by cells was dependent on CPPs’ chirality. Similarly, results of another study using dimeric fluorescence Tat (dfTat) as a model CPP revealed that inversion of CPP chirality from the L- to the D-form reduced endocytic uptake [81], as consistent with results mentioned above [80]. In a related study, a single D-amino acid substitution decreased the overall uptake of anionic CPP p28 by cancer cell lines [82]. Notably, the D-form of the abovementioned dfTat CPP exerted a prolonged cytotoxic effect, due to its persistence in the cytosol for several days, whereas the L-peptide was degraded within hours [81]. Therefore, highly cytotoxic D-peptides, such as the MUC1-C inhibitor CPP GO-203 that disrupted binding of MUC1-C to BAX in vitro, solely comprise D-amino acids and thus have been found to effectively combat tumors [83] and induce death of lung cells [84] and prostate cancer cells [85]. Examples of CPPs containing D-amino acid are shown in Table 3.

**Table 3 pharmaceutics-15-02093-t003:** Examples of CPPs containing D-amino acid.

Name	Peptide Sequence	Ref.
D-KLA	klalklalkalkaalkla	[86]
D-pVEC	lliilrrrirkqahahsk	[72]
D-SAP	(vrlppp)_3_	[77]
D-Tat	rkkrrqrrr	[73]
D-Penetratin	rqikiwfqnrrmkwkk	[75]
GO-203	cqcrrkn	[83,84,85]
Oligoarginine	rrrrrrrr	[74]
D-R_9_	rrrrrrrrr	[80]
dL1-p28	lSTAADMQGVVTDGMASGLDKDYLKPDD	[82]
dL24-p28	LSTAADMQGVVTDGMASGlDKDYLKPDD	[82]
dD28-p28	LSTAADMQGVVTDGMASGLDKDYLKPDd	[82]
dD22-p28	LSTAADMQGVVTDGMASGLDKdYLKPDD	[82]
D-hLF	qwqrnmrkvr	[80,87]
MUC2	tpTPTGTQtpt	[78]
D-dfTAT	(ck(TMR)rkkrrkrrrg)_2_	[88]
RICK	kwllrwlsrllrwlarwlg	[76]
D-Cady-k	glwralwrllrslwrllwk	[76]

Lowercase letters represent D-form amino acids, and the capital letters represent L-form acids.

## 4. CPPs as Vectors for Cellular Delivery of Peptides

After the discovery of the first CPP, the Tat peptide (GRKKRRQRRRPPQ), additional CPPs were discovered, such as Pep-1 (KETWWETWWTEWSQPKKKRKV), polyarginine, and penetratin (RQIKIWFQNRRMKWKK). Indeed, as shown in Figure 5, these CPPs are the most commonly used membrane-permeable vectors, due to their abilities to cross epithelial cell membranes and other physiologic barriers, such as the blood–brain barrier (BBB). The BBB selectively protects the central nervous system (CNS) from external insults, but its function can limit the passage of therapeutic molecules. CPPs are also beginning to be investigated for use as carriers across the blood–brain barrier. Researchers have validated the ability of CPPs to cross the blood–brain barrier using cell model simulations. A peptide, SUV-RF, containing a single alpha-helix, was bound to liposomes encapsulating gefitinib in the hopes of treating non-small cell lung cancer cells that have metastasized to the brain. Using the bEnd.3 cell model to simulate the blood–brain barrier, lung adenocarcinoma PC9 cells were grown with a Transwell insert inserted underneath, and it was shown that liposomes containing the membrane-penetrating peptide significantly reduced PC9 cell viability [89]. The ability of CPPs to cross cell menbranes does not correlate well with the ability to cross the blood–brain barrier. Spiegeler evaluated the ability of five cell-penetrating peptides, pVEC (LLIILRRRIRKQAHAHSK), SynB3 (RRLSYSRRRF), Tat11 (YGRKKRRQRRR), TP10 (AGYLLGKINLKALAALAKKIL), and TP10-2 (AGYLLGKINLKALAALAKKIL) to cross the BBB and found that pVEC, SynB3, and Tat11 had higher intracerebral influx rates, whereas those of TP10 and TP10-2 were relatively low. Co-injected radiolabeled BSAs were not detectable in the brain, so their crossing of the blood–brain barrier was not achieved by increasing the permeability of the blood–brain barrier. pVEC had the highest brain influx rate and its brain efflux rate was the lowest, suggesting that the blood–brain barrier is selective for the permeation of membrane-penetrating peptides [90].

To better evaluate the ability of CPPs to promote the crossing of active substances across the blood–brain barrier, Barra et al. constructed the fluid dynamic BBB model and demonstrated that gh625-functionalized liposomes enhanced pituitary adenylate cyclase-activating polypeptide (PACAP) delivery across the BBB [91]. Currently, CPPs are used to transport peptides into cells that perform various therapeutic tasks to alleviate various diseases, such as cancer, diabetes, bacillosis, and others [92,93].

### 4.1. Anticancer Peptides (ACPs)

To date, many anticancer peptides, known as “cargos,” have been fused to CPPs that have promoted their delivery into tumor cells to treat cancer. For example, p16Ink-penetratin [94], p27kip-Tat [95], and p21WAF1/CIP1-Tat [96] (Table 4), which are composed of CPPs linked to peptides derived from natural protein inhibitors of cyclin-dependent kinases (CDKs), have been reported to inhibit tumor growth in vitro and in vivo. Meanwhile, p53C’, an anticancer peptide derived from the C-terminus of p53, has been conjugated to both the CPP D-isomer FHV (rrrrnrtrrnrrrvr) and D-Pas (fflipkg). The resulting ACP facilitated cellular uptake and dose-dependently inhibited growth of glioma-initiating cells (GICs) in 2D monolayer cultures, 3D multi-cellular spheroids, and xenografted animal models [97]. Moreover, linkage of the p53C’ D-isomer peptide to influenza virus hemagglutinin-2 protein (riHA2, gdimgewgneifgaiagflgc) via a disulfide bridge resulted in creation of the chimeric peptide d11R-p53C-riHA2, which inhibited bladder cancer cell proliferation while promoting apoptosis. In animal studies, this peptide prolonged survival of mice with transplanted tumors as compared to untreated controls [98]. Another ACP, which consisted of PNC-28 (a peptide derived from the MDM-2-binding domain of p53) conjugated to penetratin (RQIKIWFQNRRMKWKK) via its carboxyl terminal end, blocked pancreatic cancer cell growth in vivo and induced tumor cell necrosis expression in 13 different human cancer cell lines without affecting normal cells (i.e., pancreatic acinar cells, breast epithelial cells, and human stem cells) [99,100]. Furthermore, the 14-residue APC KLA (KLAKLAKKLAKLAK) and its D-isomer analog (klaklakklaklak) (Table 4), which are typical cationic and amphiphilic α-helical proapoptotic peptides, destroyed mitochondrial membranes after they were internalized by cancer cells. In addition, conjugation of KLA with various CPPs, such as PTD-5 (GYGRKKRRQRR) [101], Tat, and polyarginine, exhibited greater uptake efficiency, lower cytotoxicity, and greater antitumor activity than unconjugated KLA. Similarly, when KLA was conjugated with IMT-P8 (RRWRRWNRFNRRRCR), a novel CPP with transdermal delivery capability, the resulting IMT-P8-KLA was able to penetrate into the skin in vivo and gain entry into cells of various in vitro-cultured cell lines, resulting in significant cell death [102]. Another ACP, the cyclic peptide P15 (CWMSPRHLGTC), prevented a serine–threonine kinase (casein kinase 2, CK2) that is frequently dysregulated in many human tumors from phosphorylating its substrate, resulting in antitumor effects both in vitro and in vivo. In fact, when P15 was fused to Tat, the resulting P15-Tat ACP induced apoptosis by activating caspase and cellular cytotoxicity toward various tumor cell lines (Table 4). Additionally, P15-Tat exhibited antitumor properties when it was directly injected into a murine tumor model [103]. More recently, tumor necrosis factor (TNF)-related apoptosis-inducing ligand (TRAIL) has been shown to selectively induce apoptosis in cancer cells. To improve TRAIL antitumor and cytocidal effects against pancreatic cancer cells, the 114–121 amino acid coding sequence “VRERGPQR” of TRAIL was substituted with the amino acid sequence “RRRRRRRR” to create the novel mutant TRAIL protein TRAIL-Mu3, which induced pancreatic cancer cell death more efficiently than TRAIL by up-regulating DR5 and activating caspase [104]. In addition to the abovementioned ACPs, numerous other anticancer peptides delivered by CPPs have been evaluated for anticancer activities. For example, a cationic anticancer peptide (HPRP-A1) was chemically conjugated to a tumor-homing/penetration peptide (i RGD, CRGDKGPDC) to increase its cell specificity, penetration ability, and tumor tissue accumulation as an effective strategy for improving ACP targeting [105] (Table 4). Moreover, an ACP formed by conjugation of Tat to the 17-residue segment of S100A1 protein (Table 4), a member of the human S100 family, activated normal functions of p53 to decrease cell proliferation activity and induce cell apoptosis in cancer cells [106]. In conclusion, ACPs generated through conjugation of CPPs with ACPs are more likely to enter tumor cells and exert stronger anticancer activities than unconjugated ACPs.

### 4.2. Antidiabetic Peptides (ADPs)

Diabetes is a metabolic disorder characterized by higher than normal blood glucose levels that may be treatable with ADPs, due to their high-level specificity and potency that make them promising candidates for use as type 2 diabetes mellitus (T2DM) treatments. For example, glucagon-like peptide-1 (GLP-1), an important incretin hormone composed of 30-amino acid long peptides that are produced in the intestine, can induce insulin secretion from pancreatic islets to regulate glucose homeostasis in vivo [107]. Similarly, exenedin-4 is a clinically available antidiabetic peptide derived from salivary secretions of the Gila monster (Heloderma suspectum) which shares 53% amino acid sequence homology with GLP-1 [108]. Due to the success of exendin-4, researchers have searched for new potent ADPs, resulting in recent discoveries of ADPs that can be administered via oral and nasal routes as alternative treatments to invasive ADP injections. However, many critical challenges must be overcome before effective and convenient non-invasive ADP delivery systems can be developed, including difficulties related to inefficient ADP egress from the blood circulation into the target tissue (e.g., pancreas) and ADP proteolytic degradation in vivo. In addition, methods to promote efficient ADP internalization by beta cells are needed to enable ADP triggering of intracellular insulinotropic pathways [109].

With the development of CPPs, drug delivery systems based on these peptides are increasingly being viewed as attractive tools for use in overcoming current limitations of orally administered ADPs, as shown in Table 4. The aforementioned drug delivery systems are based on two approaches that can enhance cellular absorption of ADPs: CPP conjugation to cargos and co-administration of CPPs with therapeutic molecules [110]. For example, after co-administration of CPP D-R_8_ (rrrrrrrr) and GLP-1, intestinal absorption of GLP-1 was significantly increased, while no enhancement of exenedin-4 intestinal absorption was observed. These results suggest that negatively charged GLP-1 can efficiently bind to positively charged D-R_8_ through an intermolecular interaction [111]. Nasal administration, another non-invasive drug delivery approach, may enable direct drug delivery into the systemic circulation, while avoiding hepatic first-pass metabolic reduction of drug bioavailability. Importantly, CPPs L-penetratin, D-penetratin, shuffled RK-fixed analog [shuffle (R,K fix) 2] (RWFKIQMQIRRWKNKK), and protein transduction domain (PTD) double mutants TCTP-PTD 13M_2_ and 13M_3_ have been shown to markedly enhance nasal absorption of therapeutic peptides exendin-4 and GLP-1 when CPPs and therapeutic peptides were co-administered nasally in a murine diabetes model [112,113,114].

Recently, researchers have found that TCTP-PTD 13M_2_ (MIIFRLLASHKK) reduced blood glucose levels in a diabetic mouse model by 43.3% and 18.6% as compared to glucose reductions induced by exendin-4 alone and co-administered exendin-4 plus TCTP-PTD 13 (MIIFRALISHKK), respectively. Thus, these results demonstrated that nasal co-administration of TCTP-PTD 13M_2_ with exendin-4 reduced blood glucose levels in diabetic mice more effectively than co-administration of exendin-4 plus wild-type TCTP-PTD 13 [114]. In addition, they found that simple mixing of TCTP-PTD 13M_2_ with peptide/protein drugs was a potentially viable approach for achieving effective intranasal delivery of CCPs and therapeutic peptides into animals. To test their proposed strategy, the C-terminus of exendin-4 was covalently linked to TCTP-PTD 13M_2_ then the conjugated CCP–peptide was assessed for its hypoglycemic effect in diabetic mice after subcutaneous or intranasal delivery. They found that subcutaneous injection of the conjugated peptide reduced blood glucose levels by 42.2% as compared to the 0% reduction observed in the control group, while the conjugated peptide did not produce the same effect when delivered via the nasal route [114]. By contrast, results of another study demonstrated that co-administration of exendin-4 and GLP-1 via the nasal route was more effective in reducing blood glucose levels than oral co-administration of these peptides [112]. However, D-penetratin did not exhibit any effect on nasal delivery of exendin-4 in diabetic mice [65]. Interestingly, numerous ADPs have been found to possess CPP-like properties. For example, [L28K] esculentin-2Cha (GFSSIFRGVAKFASKGLGKDLAKLGVDKVA) and esculentin-2Cha (1–30) (GFSSIFRGVAKFASKGLGKDLAKLGVDLVA) peptides derived from esculentin-2Cha, a 37-mer bioactive peptide isolated from a toxin present in frog skin secretions, exhibited insulinotropic action that was likely dependent on their cell-penetrating abilities, which supported their efficient internalization by β cells that enabled them to stimulate β cell insulin exocytosis [115,116]. Moreover, GLP-1 helix constraints and amino acid substitutions (Tyr16, Ala22) may have increased GLP-1 cationicity or hydrophobicity to enhance its penetration into cytomembranes of β cells, thus enabling it to stimulate insulin secretion [117].

### 4.3. Antimicrobial Peptides (AMPs)

AMPs, which are also referred to as natural antibiotics, have numerous functions in various animals, plants, and bacteria and thus hold promise as alternatives to antibiotics, especially for use in eradicating intracellular pathogens [118]. Importantly, CPP–AMP conjugates can cross host cell plasma membranes and bind to Gram-negative bacterial DNA and cell surface LPS to exert antibacterial effects, while also eliminating bacteria present within vacuoles and the cytosol by triggering certain signaling pathways that promote increased intracellular levels of TNF-α, IL-1β, and IL-10 [119]. Thus, conjugation of CPPs to AMPs may be an effective approach for developing novel antimicrobial agents.

Another AMP, KR-12 (which comprises residues 18–29 of human cathelicidin LL-37) (Table 4) has anti-inflammatory properties and good cell selectivity but does not efficiently penetrate the cell membrane. To address this issue, several researchers have utilized CPP Tat as a vector to facilitate KR-12 entry into osteoblasts and macrophages of patients that were postoperatively infected with S. aureus in order to kill the bacteria with minimal host cell toxicity. Moreover, Tat-KR-12 also significantly inhibited expression of NO, TNF-α, and IL-1β in LPS-stimulated RAW264.7 cells. Taken together, these results indicate that Tat-KR-12 holds promise as a new antimicrobial for treating intracellular antibiotic-resistant bacterial infections [118,120]. Meanwhile, a related CPP, CPP-bLFcin6(RRWQWR)/Tat11, was shown to enhance marine peptide N_2_-associated antibacterial activity against S. typhimurium and improve N_2_ peptide stability in macrophages when the CPP was conjugated to the C-terminus of the N_2_ peptide [121]. In related work, Lee et al. found that as compared to AMPs (magainin and M_15_), CPP(R_9_)–AMP (magainin or M_15_) conjugates exhibited significantly enhanced antimicrobial activity against Gram-negative bacteria. Notably, magainin, M_15_, and CPP–AMP all possessed weak hemolytic activity, while conjugation of AMP to the CPP did not increase AMP cytotoxicity [122].

Interestingly, peptides such as salusin-β [123], which was previously shown to alleviate cardiovascular diseases, and HEXIM1 BR, which was previously shown to kill cells of several cancer cell lines, have also be found to possess appreciable antibacterial activities after conjugation with CPPs. For example, as shown in Table 4, conjugation of salusin-β with CPP-Tat (49–57) created a peptide with potent antibacterial activities against both Gram-positive and Gram-negative bacteria [124]. Similarly, conjugation of HEXIMI BR with Pen-BR (RQIKIWFQNRRWGGQLGKKKHRRRPSKKKRHW) or Pen-RRR (RQIKIWFQNRRWGGQLGRRRHRRRPSRRRRHW) created peptides with strong bactericidal abilities against drug-resistant bacteria without harming eukaryotic cells [125].

In general, CPPs can contribute to antibacterial activities in two ways: CPPs can either serve as vectors to deliver other antibacterial agents into cytomembranes, or CPPs can directly act as antibiotic agents. Some CPPs with antimicrobial activities are referred to as antimicrobial cell-penetrating peptides (ACPPs), due to their biofilm-disrupting antimicrobial activities and similar structures, while other CPPs can be converted into AMPs through amino acid substitution [126]. For example, the chimeric peptide Pep-1 (KETWWETWWTEWSQPKKKRKV) (Table 4) has been commercialized as a non-cytotoxic CPP that can deliver protein, peptide, and antibody biomacromolecular cargoes to cells by enabling peptide cell membrane penetration, while also exerting weak antibacterial activity against Bacillus subtilis [127]. In addition, other researchers have found that Pep-1 could inhibit intracellular, but not extracellular, Chlamydia activities [128]. Notably, Pep-1-K (KKTWWKTWWTKWSQPKKKRKV) (Table 4), a new AMP derived from Pep-1, has been reported to exert stronger antimicrobial effects than Pep-1, due to its stronger cationic properties resulting from incorporation in Pep-1-K of three Glu residues in place of Lys residues, which increased the affinity of Pep-1-K peptide for bacterial membranes [129].

**Table 4 pharmaceutics-15-02093-t004:** Therapeutic peptides in this article delivered by CPPs.

Type	Therapeutic Peptide	CPP	Functions	Ref.
Anticancer peptides (ACPs)	p16Ink	Penetratin	Inhibits pancreatic cancer growth and prolongs survival in vivo	[94]
	p27kip	Tat	Inhibitor of cyclin-dependent kinases (CDKs)	[95]
	p21(WAF1/CIP1)	Tat	A cytotoxic peptide mimic of the cyclin-dependent kinase inhibitor	[96]
	p53C’	D-isomer FHV/D-Pas	An anticancer peptide derived from the C-terminus of p53 that inhibits GIC growth in vitro and in vivo	[97]
	PNC-28	Penetratin	A peptide derived from the MDM-2-binding domain of p53 that can block pancreatic cancer cell growth in vivo and induce tumor cell necrosis in 13 different human cancer cell lines	[99,100]
	KLA	PTD-5/IMT-P8/Tat/polyarginine	A proapoptotic peptide and used as a therapeutic peptide to destroy the mitochondrial membrane	[101,102]
	Cyclic peptide P15	Tat	A cyclic peptide that blocks CK2 that is frequently dysregulated in many human tumors and exhibits antitumor effect in vivo	[103]
	HPRP-A1	iRGD	A cationic peptide that kills cancer cells by disrupting the cell membrane and inducing apoptosis in vitro	[130]
	S100A1	Tat	It can influence the p53–MDM2 interaction credibly and possibly reactivates the wild-type p53 pathway	[106]
Antidiabetic peptides (ADPs)	GLP-1	D-R_8_ /penetratin/ D-penetratin /TCTP-PTD 13M_2_/TCTP-PTD 13M_3_	An important incretin hormone, derived from intestine, that can induce insulin secretion from pancreatic islets to regulate glucose homeostasis in vivo	[113,114,131]
	Exenedin-4	Penetratin/D-penetratin/TCTP-PTD 13M_2/_TCTP-PTD 13M_3_	A clinically available antidiabetic peptide derived from the salivary secretions of the Gila monster (Heloderma suspectum)	[111]
Antimicrobial peptides (AMPs)	KR-12	Tat	Residues 18–29 of human cathelicidin LL-37 have anti-inflammatory properties and good cell selectivity	[118,120]
	Magainin and M15	R_9_	Cationic and amphipathic α-helical peptides can exert their activity by permeabilizing cytoplasmic membranes	[132]
	Salusin-β	Tat/Pen-BR/Pen-RRR/HEXIM1 BR	A peptide previously used in cardiovascular diseases and in several cancer cell lines that has shown antibacterial ability after conjugation with CPPs	[133]
	Pep-1	/	A peptide that has weak activity against Bacillus subtilis and can inhibit intracellular chlamydial infection not extracellular chlamydiae	[127]
	Pep-1-K	/	A new AMP derived from Pep-1 that has stronger antimicrobial effects because of the high affinity to bacterial membranes	[129]

## 5. CPPs as Vectors for Protein Delivery to Cells

Shuttling of exogenous proteins into cells is a promising approach for achieving various therapeutic goals, such as regulation of various signaling pathways, replenishment of enzymes, etc. However, proteins are large biomacromolecules that cannot directly penetrate the cell plasma membrane to enter cells. Nevertheless, to date many covalent CPP fusion proteins and non-covalent CPP–protein conjugates have been developed that can successfully deliver proteins into cells both in vitro and in vivo (Figure 5). In the following section, as shown in Table 5, various typical therapeutic proteins, such as tumor suppressor proteins, antioxidant proteins, and antidiabetic proteins, are discussed.

### 5.1. p53 Protein

p53 (Table 5) is a tumor suppressor protein and proapoptotic protein that can induce cellular growth arrest and apoptosis in response to stress. Notably, mutations of the gene encoding p53 are responsible for almost all types of human cancers, which suggests that restoration of wild-type p53 functions may be a particularly promising cancer treatment strategy. Towards this end, several research groups have reported that a synthetic 22-mer peptide derived from the p53 C-terminus (p53C) could bind to sites within the mutant p53 protein core domain and C-terminal domain to thereby restore mutant p53 transactivation function. The peptide acted by stabilizing the core domain structure and/or establishing novel DNA contacts to trigger p53-dependent tumor cell apoptosis [134]. Nevertheless, due to the fact that these peptides are macromolecules with non-polar characteristics and molecular weights exceeding 500 Da, they cannot easily traverse the cell membrane to exert their functions [93]. Consequently, many CPPs have been employed to directly introduce p53 or the p53 mimics into malignant cells to successfully recover p53 function that can halt proliferation of many different types of cancer cells [135]. For example, a mutation of the p53 protein-encoding TP53 gene is found in most urothelial carcinomas of the bladder (UCB). To restore normal p53 function to these cells, researchers have successfully used the cell-penetrating peptide polyarginine (R_11_, RRRRRRRRRRR) as a vehicle to deliver functional p53C into UCB cells. Their results revealed that after R_11_-p53C treatment, transcription levels of several p53 target genes were up-regulated significantly and growth of UCB cells was inhibited by activation of the p53-dependent pathway [136]. Meanwhile, R_11_ has been observed to deliver functional p53 across cell plasma membranes of human oral cancer cells, resulting in induction of p21/WAF promoter activity and tumor cell apoptosis in cells with mutations of the TP53 gene without affecting normal cells [136,137]. Moreover, R_11_-p53C not only inhibited bladder tumor growth in vivo, but also dramatically extended survival of animals in in vivo orthotopic and lung metastatic cancer animal models. Based on the abovementioned results, R_11_-p53C may be a promising therapeutic drug for use in treating UCB cases, with special significance for patients with metastatic disease [136].

Several studies have shown that polyarginine that is covalently linked to p53 protein is sequestered within the cytoplasm and thus can bind via its NH_2_-terminal domain to the pH-dependent fusion peptide of the influenza virus hemagglutinin-2 subunit (HA2) to promote its efficient transport into glioma cell nuclei [138]. Nonetheless, despite attempts made by numerous researchers to create drugs that can directly modulate the activity of the p53 transcription factor, no drugs with this activity have been approved for clinical use to date [139]. However, several candidates are being clinically evaluated, such as CPP p28, a promising drug derived from the Pseudomonas aeruginosa azurin protein. In fact, evaluation of CPP p28 in a Phase I clinical trial is complete and the drug and is currently being evaluated in a Phase II trial to determine the optimal dose of the drug when used as an antitumor treatment [12].

### 5.2. Tumor Necrosis Factor-Related Apoptosis-Inducing Ligand (TRAIL)

TRAIL (Table 5), a key death receptor ligand involved in extrinsic apoptotic signal routing, can selectively induce apoptosis in cancer cells without harming normal cells. Thus, TRAIL holds promise as a therapeutic anticancer agent. However, many primary tumors and cancer cell lines have been found to be resistant to TRAIL-induced apoptosis, prompting researchers to conduct studies to solve this problem. In one such study, a novel mutant protein TRAIL-Mu3 was synthesized by replacing the TRAIL 114–121 amino acid sequence “VRERGPQR” with the CPP sequence “RRRRRRRR.” The results of the study indicated that addition of TRAIL-Mu3 to pancreatic cancer cells resulted in penetration of the cell plasma membrane by TRAIL-Mu3, which resulted in more extensive pancreatic cancer cell death than that induced by wild-type TRAIL. These results also suggest that the cell-penetrating sequence increased the binding affinity between TRAIL and the cell membrane and up-regulated DR5 to activate caspase [104]. Furthermore, the TRAIL mutant R6 (MuR6-TR) exerted a greater inhibitory effect on tumor growth than did natural TRAIL when used to inhibit activities of pancreatic carcinoma cell lines BxPC-3 and PANC-1, thus indicating that the CPP sequence may have enhanced the ability of TRAIL-Mu3 to penetrate the cell membrane without increasing cell sensitivity to TRAIL [140].

Several studies have shown that CPPs can indirectly enhance the antitumor effect of TRAIL. For example, a DNA-binding CPP peptide (YGRKKRRQRRR-G_3_-R_9_) composed of a cell-penetrating domain and a DNA-binding domain delivered the TRAIL-encoding gene into adipose tissue-derived mesenchymal stem cells (ASCs) to enhance expression of TRAIL and inhibit glioma cell proliferation in vitro and in vivo [141]. Meanwhile, studies of BH_3_-interacting domain death agonist (BID) protein, an integral member of pathways related to death receptor functions and mitochondrial cell death-inducing processes, may have relevance to TRAIL’s role in the extrinsic apoptotic signaling pathway. For example, silencing of BID expression renders tumor cells resistant to death ligand-induced apoptosis, thus implying that overexpression of BID may make tumor cells more sensitive to TRAIL. When Tat CPP is fused with BID, it should be delivered to cancer cells in a controlled manner that would likely enhance TRAIL efficacy, as consistent with results of a study showing that Tat CPP could efficiently sensitize prostate PC3 cells and non-small human lung cancer A549 cells to TRAIL [142].

### 5.3. Superoxide Dismutase (SOD)

SOD (Table 5) is a well-characterized essential antioxidant enzyme that can scavenge oxygen free radicals and inhibit lipid peroxidation to protect cells from detrimental effects of various oxidant compounds [143]. SOD includes three types of molecules, namely SOD1, SOD2, and SOD3, which are present in mammalian cells. Among them, SOD1 is mainly present in the cytoplasm, although a small amount of the enzyme is found within the mitochondrial intermembrane space and the nucleus; SOD2 is present within the mitochondrial matrix; SOD3 is present within the extracellular space [144]. SOD1 and SOD3 are Cu/Zn-SODs, since they contain copper and zinc in their active sites, while SOD2 is a Mn-SOD, since manganese ions are present in its active site [145]. Notably, it has been reported that diseases such as cancers, neurodegenerative diseases, and diabetes are related to antioxidant enzyme deficiencies, thus implying that exogenous delivery of antioxidant enzymes may be a promising method to treat diseases related to oxidative stress [146]. However, due to its large size and limited ability to cross biological membranes, exogenous SOD does not readily penetrate cell membranes to gain entry into cells and tissues to protect them from oxidative damage. To address this issue, various CPPs, such as Tat, Pep-1, and R_9_, have been developed for use as vectors to deliver exogenous SOD into cells. For example, Tat-SOD fusion protein, which has been transduced into pancreatic beta cells of diabetes model mice to protect them from destruction, has also been shown to induce protective effects against neuronal cell death and ischemic insults in mice after peritoneal injection of the fusion protein [26]. In addition, Tat-SOD1 has been found to act as a radioprotector, since it exhibited a strong radioprotective effect that led to improved growth rates of mice with irradiation-induced lung damage [147]. Furthermore, Tat-SODs have been observed to exert anticancer effects, whereby Tat-Cu/Zn-SOD treatment of tumor-ridden rats attenuated bone cancer pain (BCP) and relieved side effects of chemotherapy [148]. Moreover, Tat-Mn-SOD inhibited proliferation of hepatoma cells (HepG2) by reducing intracellular ROS levels without affecting normal hepatocytes (QSG-7701) [149].

The short amphipathic CPP, Pep-1 (KETWWETWWTEWSQPKKKRKV), which was designed to serve as a peptide/protein carrier, has been shown to provide several advantages, such as stability in physiological buffer and serum and lack of toxicity [150]. Another Pep-1-SOD fusion protein has been demonstrated to protect neuronal cells in a cell-based in vitro model and an animal model of Parkinson disease (PD) induced by paraquat in vitro and in vivo, respectively [151]. Moreover, Pep-1-Cu and Zn-SOD also exerted neuroprotective effects against ischemic neuronal damage in a rabbit spinal cord injury model and strongly protected adipose-derived mesenchymal stem cells from ischemic damage by maintaining a balance between lipid peroxidation and antioxidant activities [152,153]. When fused with Pep-1, SOD has been shown to protect the heart by penetrating the myocardium to reduce superoxide anion and hydrogen peroxide levels to mitigate myocardial ischemia–reperfusion-induced damage in in vivo and in vitro cardiac hypoxia–reoxygenation injury models [154,155]. In addition, results of several investigations have suggested that Pep-1-SOD1 reduced lipid peroxidation to ameliorate memory deficits while triggering cell proliferation and neuroblast differentiation, which occur at abnormally low levels in dentate gyri of D-galactose-induced aged mice [156].

In a study reported in 2000, two peptides, including a 9-mer L-arginine (R_9_) peptide and a 9-mer D-arginine (r_9_) peptide, were designed, synthesized, and then assessed for cellular uptake efficiency and other characteristics. R_9_ exhibited a more efficient cellular uptake rate than Tat (49–57) (20-fold), as was determined by Michaelis–Menton kinetic analysis, while r_9_ exhibited >100-fold greater uptake efficiency than Tat (49–57). Moreover, R_9_ and r_9_ offered additional advantages, such as protease resistance, greater accessibility, and lower production costs [73]. Fusion of hMnSOD with R_9_ resulted in creation of hMnSOD–R, which could inhibit proliferation of various cancer cell lines in a dose-dependent manner. In addition, hMnSOD–R_9_ induced HeLa cell apoptosis by up-regulating cleaved caspase-3 and down-regulating phospho-STAT3 pathway activation, while also inducing cell cycle arrest in a dose-dependent manner that caused the cells to remain in sub-G_0_ phase [157].

It is worth noting that although external SOD treatment may inhibit TNF-α-induced endothelial cell production of superoxide anion, it cannot inhibit endothelial cell migration. However, Tat-SOD, which was produced by coupling Tat to SOD, blocked SOD transport between monocytes and endothelial cells to thereby protect endothelial cells from oxidation-driven, inflammation-induced atherosclerosis [158]. Taken together, the abovementioned studies showed that CPPs-SOD possessed markedly greater therapeutic potential, stronger antioxidant activity, and greater therapeutic efficiency than SOD [159].

### 5.4. Monoclonal Antibody

Monoclonal antibodies are immunoglobulins that can target specific antigenic epitopes and thus hold promise as powerful immunotherapeutic tools for use in oncological investigations. Currently more than a dozen antibody-based drugs have been approved by the FDA for the treatment of cancer. However, the large size of these proteins (150 kDa) prevents them from passing through the cell membrane, prompting researchers to generate various CPPs to shuttle antitumor antibodies into specific types of living cells [160]. For example, CPPs Pep-1 and PEPth have been shown to significantly improve antibody cytosolic penetration, particularly when they are fused to antibodies above or below antibody hinge regions [161]. Once internalized, antibodies can target disease-relevant human proteins present within the intracellular environment to modulate activities of hitherto untargetable pathways through protein–protein interactions. Nonetheless, another obstacle that has limited the use of such antibodies for this purpose has been their susceptibility to endosomal entrapment, prompting one group of researchers to design trimeric CPPs to deliver functional immunoglobulin G antibodies and Fab fragments into cellular cytosolic and nuclear compartments to act on intracellular targets. Notably, the four trimers that were designed based on linear and cyclic sequences of archetypal CPP Tat were more effective than monomers in blocking activities of selected targets, due to their abilities to directly escape from vesicle-like bodies when present intracellularly at therapeutically effective concentrations [63].

Cell-penetrating antibodies, such as RT11 and its variant equipped with an additional tumor-associated integrin binding moiety, have also been reported to be internalized by living tumor cells and to enter the cytosol, where they selectively inhibit cell proliferation by targeting oncogenic Ras mutants in vitro and in vivo [162]. Nevertheless, even when monoclonal antibodies are conjugated to CPPs, monoclonal antibody targeting of intracellular molecules is still influenced by other factors, such as low solubility, rapid elimination, and intracellular stability. However, blocking of intracellular target activities using monoclonal antibodies encapsulated in nanosystems is a promising strategy that may overcome these challenges. For example, the actin cytoskeleton protein CapG is usually overexpressed in breast cancer cells, where it appears to play a role in tumor cell dissemination and metastasis. Notably, treatment of breast cancer cells with a CapG-specific nanobody conjugated to various CPPs effectively reduced breast cancer metastasis by blocking CapG activity [163].

### 5.5. Insulin

CPPs have been studied for several years as oral absorption enhancers for antidiabetes biomacromolecular therapeutics, such as insulin and other antidiabetic peptides, since CPPs can increase biomacromolecule intracellular delivery efficiency and reduce biomacromolecule degradation in the harsh gastrointestinal environment [164]. For example, when insulin was linked to Tat to generate Tat-insulin (Table 5), the conjugated peptide passed through a Caco-2 cell monolayer intact and exhibited 7-fold greater intestinal absorption efficiency than insulin [165]. Moreover, co-administration of insulin and CPPs such as penetratin, pVEC (LLIILRRRIRKQAHAHSK), RRL helix (RRLRRLLRRLRRLLRRLR), D-forms of R_8_, and L-forms of R_8_ significantly enhanced the intestinal absorption efficiency of insulin [111,166]. Furthermore, D-forms of R_8_ only improved intestinal absorption of negatively charged macromolecules (e.g., insulin, gastrin, and GLP1), due to electrostatic interactions between D-R_8_ and theses cargoes [111].

Meanwhile, researchers have devised other strategies to enhance CPPs’ applicability as efficient vectors to improve insulin absorption. For example, results of several studies have shown that the shuffled RK-fixed analog [(shuffle (R,K-fix)-2] (RWFKIQMQIRRWKNKK) when co-administered with insulin improved intestinal absorption of insulin more significantly than parent molecules penetratin and SAR_6_EW (C_17_H_35_-R_6_EW). Using another approach, an amphiphilic lipopeptide variant of R_8_ was developed and found to enhance intestinal permeability of insulin more effectively than other R_8_ variants both in vitro and in vivo (Table 5) [167,168].

To date, increasingly advanced and novel strategies have been used to improve CPPs’ delivery efficiencies. For example, one such strategy entailed the development of NPs consisting of functionalized poly(arginine)_8_ enantiomers (L-R_8_ and D-R_8_) conjugated to poly(lactic-co-glycolic acid) (PLGA) for use in oral insulin delivery. Importantly, when these NPs were orally co-administered with insulin, dramatically enhanced intestinal absorption of insulin was observed in vitro and in vivo, especially for D-R_8_ [155]. Moreover, in in vitro and in vivo experiments, PLGA NPs co-modified with penetratin and an engrailed secretin peptide (sec) enhanced oral delivery of insulin to provide a more intense hypoglycemic effect than was observed for NPs modified with penetratin alone. However, NPs consisting of other sec-modified CPPs (e.g., Tat and R_8_) did not provide significantly better oral insulin delivery than was provided by Tat-NPs or R_8_-NPs [169].

Importantly, oral insulin must penetrate two main absorption barriers in order to exert a hypoglycemic effect, an enzymatic barrier and a mucosal barrier. Based on this premise, one research group designed NPs composed of insulin and a CPP, then coated them with a dissociable hydrophilic layer consisting of N-(2-hydroxypropyl) methacrylamide co-polymer (pHPMA) derivatives. As compared to free insulin, NPs provided markedly greater insulin absorption performance both in vitro and in vivo [170].

Nasal administration, another non-invasive approach for achieving absorption of therapeutic proteins and peptides, provides an additional advantage by serving as an effective drug delivery route to the brain that bypasses the BBB [171]. In order to utilize this pathway for insulin delivery, in 2008 several studies were conducted to evaluate efficacies of L- or D-penetratin and L- or D-R_8_ for nasal insulin delivery. Among them, L-penetratin most effectively promoted insulin absorption by effectively enhancing insulin transport across the nasal membrane [65]. Concurrently, studies conducted by a different research group demonstrated that CPPs could be used to improve insulin absorption by the alveolar epithelial barrier. Their CPPs consisted of insulin conjugated via disulfide bridge linkages to N-terminal residues of several cationic CPPs (e.g., Tat, oligoarginine, oligolysine). Their results indicated that CPPs containing oligoarginine may be used to increase alveolar insulin absorption rates, while also demonstrating that covalent conjugation of insulin to r_9_ was required to maximize insulin transport efficiency [172].

**Table 5 pharmaceutics-15-02093-t005:** Representative therapeutic proteins delivered by CPP.

Therapeutic Protein	CPPs	Function	Ref.
P53 protein/p53C	Polyarginine, R_11_/P28	A tumor suppressor protein and proapoptotic protein that can induce growth arrest and apoptosis in response to cellular stress	[12,136,137]
Tumor necrosis factor-related apoptosis-inducing ligand (TRAIL)	TRAIL mutant R_6_	One of the death receptor ligands in extrinsic apoptotic signal routing that can selectively induce apoptosis in cancer cells while avoiding normal cells	[140]
Superoxide dismutase (SOD)	Tat/Pep-1/R_9_/r_9_	It is an essential antioxidant enzyme that can scavenge oxygen free radicals and inhibit lipid peroxidation to protect cells to avoid various oxidant compounds	[26,147,148,149,150,151,152,153,154,155,156,157,158,159]
Immunoglobulin-G	Tricyclic Tat (Figure 3A)	IgG is the most abundant antibody that plays a critical role in linking the adaptive immune response to the innate immune system	[63,173]
Insulin	Tat/penetratin/pVEC/r_8_/R_8_	An endogenous protein known to lower blood sugar in diabetics and improve cognition and learning by stimulating its receptor in the brain	[111,166]

## 6. Conclusions

CPPs hold promise as delivery vehicles of biomacromolecules such as peptides and proteins and thus will likely play an important role in the pharmaceutical field. In this review, we describe natural CPPs obtained from animal venom, as well as different generations of synthetic CPPs, such as cyclic CPPs, glycosylated CPPs, and D-type CPPs, that were designed and generated via different techniques. Importantly, both natural and artificially produced CPPs possess characteristics of high transport efficiency and high proteolytic stability that make them promising drug delivery vehicles, including CPPs that deliver peptide/protein drug treatments for various diseases that are the focus of this review. Although CPPs show great promise as drug delivery vehicles that can directly transport biomacromolecules into cells for in vitro and in vivo applications, their powerful membrane-penetrating effect is a double-edged sword that may create new challenges. For example, due to their lack of target cell penetration specificity, systemic administration of current CPP–drug combinations could lead to widespread drug distribution throughout the body that would increase drug concentrations in healthy organs, leading to greater risk of toxicity and associated side effects. Therefore, improving CPPs’ targeting of pathological tissues while preserving their transmembrane functions is a key challenge that must be overcome before CPPs can be administered to patients in clinical settings in the future.

## Figures and Tables

**Figure 1 pharmaceutics-15-02093-f001:**
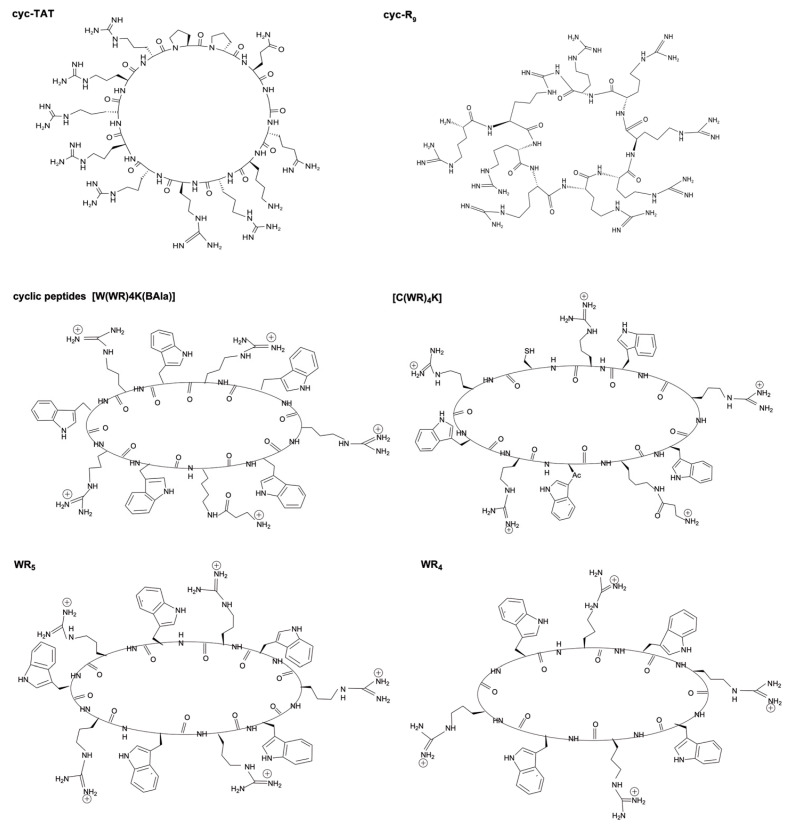
Cyclic CPPs mentioned in this article.

**Figure 2 pharmaceutics-15-02093-f002:**
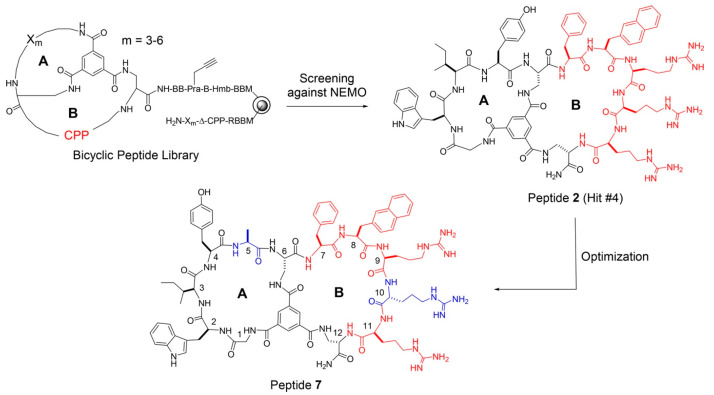
Structures of the bicyclic peptide library, peptide 2 (hit #4), and peptide 7. Amino acid residues in peptide 7 are numbered from the N- to C-terminus. The CPP sequence is shown in red, while residues modified during optimization are shown in blue. B, β -alanine; CPP, cell-penetrating peptide; Hmb, hydroxylmethylbenzoyl; Pra, propargylglycine; Δ, L-2,3-diaminopropionic acid (Dap). Reprinted with permission from [60]. Copyright 2018 American Chemical Society.

**Figure 3 pharmaceutics-15-02093-f003:**
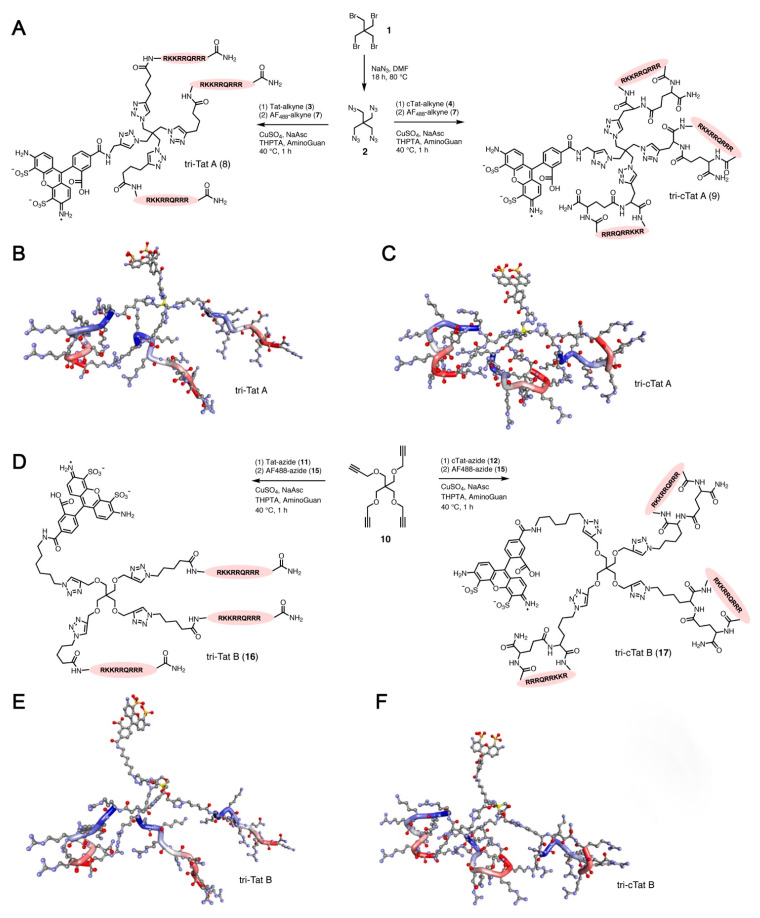
Synthesis of trimer Tat constructs. Synthesis of tri-Tat A and tri-cTat A (**A**) and synthesis of tri-Tat B and tri-cTat B (**D**). In silico-generated ball-and-stick model of linear Tat A trimer (**B**). In silico-generated ball-and-stick model of cyclic Tat A trimer (**C**). In silico-generated ball-and-stick model of linear Tat B trimer (**E**). In silico-generated ball-and-stick model of cylicTat B trimer (**F**). Reproduced with permission from reference [63].

**Figure 5 pharmaceutics-15-02093-f005:**
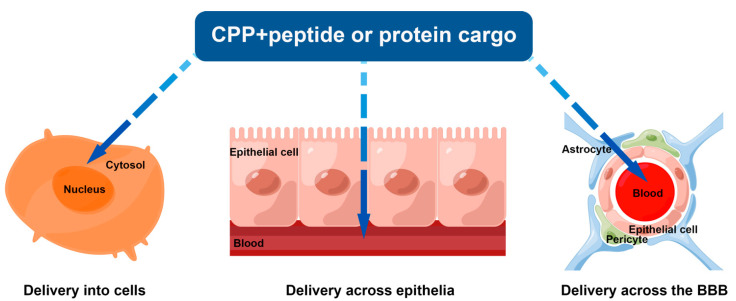
Applications of cell-penetrating peptides related to delivery of peptides and proteins into cells and across the epithelial barrier and blood–brain barrier (generated using Figdraw).

## Data Availability

Data supporting reported results can be found.

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
