# Peer review of "Recent Advances of Cell-Penetrating Peptides and Their Application as Vectors for Delivery of Peptide and Protein-Based Cargo Molecules"

_pharmaceutics, 2023, doi:10.3390/pharmaceutics15082093_

Round 1
Reviewer 1 Report
In this review the authors have summarised the recent advances in the delivery of peptides and protein cargos. The topic of the review is timely and interesting for the readers. Unfortunately, there are many older references (before 2017) in the review. Their number is more than 50% in the section 4 and 5 which should provide an overview of the recent papers about peptide and protein delivery. In my opinion more recent papers should be given at least in the last sections.
Remarks and questions:
1. It does not clear why the CPPs from venoms (Section 2) are presented in the review. I could not find any examples of them in peptide or protein delivery (Section 4 and 5).
2. In section 2.1 many times protamine is written instead of crotamine.
3. There is a short sequence from the crotamine in the Table 1. Why was this selected from the 42 amino acid long sequence?
4. In Table 1 there are just the name, the source and the reference of the sequence. Thus I do not understand „ As shown in Table 1, protamine is a versatile molecule with an immunomodulatory activity that is rapidly internalized by cells.”
5. In the line 122 the reference 17 is from 2011, so it is not „more recently”.
6. Cyclic or glycosylated peptides are not used in the given references to deliver peptides or proteins. So it is not clear why these modifications were reviewed.
7. In the Section 4 and 5 there are many abbreviations of CPPs were used for delivery. It would be worth giving their sequences.
Author Response
Dear reviewers or editors:
Hi, I would like to extend my sincere appreciation to you for valuable comments on our paper. Best regards are expressed to your rigorous attitude. Here, we have given a detailed explanation on mentioned points of the paper as follows after revisions considering suggestions of the reviewers or editors. Following is our point-by-point reply.
Response to the reviewers' comments
Reviewer 1
1.It does not clear why the CPPs from venoms (Section 2) are presented in the review. I could not find any examples of them in peptide or protein delivery (Section 4 and 5).
Response: Thanks for your question. In first part, this review summarizes the recent advance of cell penetrating peptides (CPPs) including the discovery of CPPs and the generations of CPPs. These two sections describe the interesting progress of CPPs development and provide recent highlights on CPPs for readers. Then, this paper analyzes the application of CPPs on the delivery of peptides and proteins-based cargoes in second part. Some CPPs (not all CPPs) in first part are associated with this application. However, we believe, it is possible that in future all CPPs mentioned including CPPs from venoms would display a promising application in the delivery of peptides and proteins. Your concern is helpful for us to improve the manuscript.
- In section 2.1 many times protamine is written instead of crotamine..
Response: Thanks for your suggestion. The protamine has been changed to “crotamine”.
3.There is a short sequence from the crotamine in the Table 1. Why was this selected from the 42 amino acid long sequence?
Response: Thanks for your suggestion. The 42 amino acid long sequence has been corrected to “AASSSGGPPPGGGGGCCCCCMILTPPTTLLLLLLLLLHHAATAV”.
- In Table 1 there are just the name the source and the reference of the sequence. Thus I do not understand As shown in Table 1 protamine is a versatile molecule with an immunomodulatory activity that is rapidly internalized by cells.
Response: Thanks for your question. As shown in Table 1 has been deleted.
- In the line 122 the reference 17 is from 2011 so it is not more recently.
Response: Thanks for your question. The reference from 2011 have been changed to “2018”.
6.Cyclic or glycosylated peptides are not used in the given references to deliver peptides or proteins. So it is not clear why these modifications were reviewed.
Response: Thanks for your question. As mentioned before, at first, this paper summarizes the recent advance of cell penetrating peptides (CPPs) including the discovery of CPPs and the generations of CPPs. The summary is interesting, and reflect the new advance and development trend of CPPs. However, some of the new advance abouts CPPs is not immediately applied in the delivery of peptides/proteins delivery. Your question is valuable.
7.In the Section 4 and 5 there are many abbreviations of CPPs were used for delivery. It would be worth giving their sequences.
Response: Thanks for your question. The abbreviations of CPPs in the Section 4 and 5 have been given by blue text.
Considering the suggestions of reviewers or editors, we have carried out a revision carefully. Detailed revision on manuscript has been labeled by blue text.
Thanks again for your time and effort, and for helping us to improve the manuscript.
Best regards,
Corresponding author: Nian-Qiu Shi.
Ph D, professor.
Department of Pharmaceutics, School of Pharmacy, Jilin Medical University, 5 Jilin Street, Fengman District, Jilin City, 132013, Jilin Province, China.
Tel: +86 0432 64560528
Email: shinianqiu2009@163.com.

Reviewer 2 Report
The review entitled “ Recent Advances of Cell-Penetrating Peptides and Their Application as Vectors for Delivery of Peptide and 3 Protein-based Cargo Molecules” is well-written and includes an overview of CPP from natural to synthetic sources.
I have just minor issues:
1. The name of species (such as Argentinean rattlesnakes) should be written in italic.
2. The authors describe the CPP derived from plants, insects, and other species and they do not describe the CPP derived from viruses. In the literature, there are different CPPs considered good candidates for the delivery of drugs crossing different barriers and different applications (see ref: Biochim Biophys Acta. 2015 Jan;1848(1 Pt A):16-25; Mol Plant Microbe Interact. 2011 Jan;24(1):25-36; Pharmaceutics. 2022 Jul 25;14(8):1544). I suggest the authors should be added to this section.
3. In the section “4. CPPs as vectors for cellular delivery of peptides”, the authors describe the CPP for the delivery of the cargos across the BBB. In the literature, there are many CPPs (pVEC, SynB3, Tat 47–57, transportan 10 (TP10) and TP10-2, gH625) favoring this BBB crossing (see ref: Front Physiol. 2022 Aug 19;13:932099.; PLoS One 2015 Oct 14;10(10):e0139652.) I suggest adding the section on CPPs as vectors for BBB in the review.
Author Response
Dear reviewers or editors:
Hi, I would like to extend my sincere appreciation to you for valuable comments on our paper. Best regards are expressed to your rigorous attitude. Here, we have given a detailed explanation on mentioned points of the paper as follows after revisions considering suggestions of the reviewers or editors. Following is our point-by-point reply.
Response to the reviewers' comments
Reviewer 2
1.The name of species (such as Argentinean rattlesnakes) should be written in italic.
Response: Thanks for your question. All the names of species have been changed to “italic” by blue text.
- The authors describe the CPP derived from plants insects and other species and they do not describe the CPP derived from viruses. In the literature there are different CPPs considered good candidates for the delivery of drugs crossing different barriers and different applications (see ref: Biochim Biophys Acta.2015 Jan;1848(1PtA):16-25Mol Plant Microbe Interact. 2011 Jan; 24(1):25-36; Pharmaceutics.2022 Jul 25;14(8):1544) I suggest the authors should be added to this section.
Response: Thanks for your question. The section on CPPs considered good candidates for the delivery of drugs crossing different barriers and different application have been added by blue text.
3.In the section 4. CPPs as vectors for cellular delivery of peptides, the authors describe the CPP for the delivery of the cargos across the BBB. In the literature there are many CPPs (pVEC SynB3 Tat 47-57 transportan 10(TP10) and TP10-2qH625) favoring this BBB crossing (see ref: Front Physiol 2022 Aug19; 13:932099; PLoS One 2015 Oct 14;10(10)e0139652) I suggest adding the section on CPPs as vectors for BBB in the review.
Response: Thanks for your question. The section on CPPs as vectors for BBB have been added by blue text.
Considering the suggestions of reviewers or editors, we have carried out a revision carefully. Detailed revision on manuscript has been labeled by blue text.
Thanks again for your time and effort, and for helping us to improve the manuscript.
Best regards,
Corresponding author: Nian-Qiu Shi.
Ph D, professor.
Department of Pharmaceutics, School of Pharmacy, Jilin Medical University, 5 Jilin Street, Fengman District, Jilin City, 132013, Jilin Province, China.
Tel: +86 0432 64560528
Email: shinianqiu2009@163.com.

Round 2
Reviewer 1 Report
I accepted the answers and modifications. The newly added recent references increase the value of the manuscript.